# African Swine Fever: The State of the Art in Italy

**DOI:** 10.3390/ani13192998

**Published:** 2023-09-22

**Authors:** Silvia Pavone, Carmen Iscaro, Annalisa Dettori, Francesco Feliziani

**Affiliations:** 1National Reference Laboratory for Pestivirus and Asfivirus, Istituto Zooprofilattico Sperimentale dell’Umbria e delle Marche “Togo Rosati”, 06126 Perugia, Italy; c.iscaro@izsum.it (C.I.); f.feliziani@izsum.it (F.F.); 2Regional Veterinary Epidemiology Observatory, Istituto Zooprofilattico Sperimentale dell’Umbria e delle Marche “Togo Rosati”, 06126 Perugia, Italy; a.dettori@izsum.it

**Keywords:** African swine fever virus, epidemic disease, epidemiology, exit strategy, Italy, pig, wild boar

## Abstract

**Simple Summary:**

African swine fever (ASF) is a severe viral disease of domestic pigs and Eurasian wild boars (*Sus scrofa*) that is caused by the African swine fever virus (ASFV). ASF is endemic in sub-Saharan Africa, where 24 genotypes of the virus have been reported. Between the late 1950s and the early 1980s, genotype I ASFV emerged in Europe, including Italy. In June 2007, a second ASF epidemic wave caused by genotype II was registered, involving several European and extra-European countries, including Italy in 2022. The present paper aims to provide the state of the art of ASF in Italy, describing the course of ASF in wild boars and domestic pigs as an example of multiple concurring different scenarios. Sardinia is coping with the last phase of the eradication of the disease by applying the exit strategy. Conversely, four clusters of infection located in North, Central, and South Italy are still ongoing. The unique and complex Italian experience in ASF-controlling may be useful to increase know-how on the efficacy of strategies and measures, as well as issues that could be further improved.

**Abstract:**

African swine fever (ASF) is a severe viral disease of domestic pigs and Eurasian wild boars (*Sus scrofa*) caused by the African swine fever virus (ASFV). ASF is endemic in sub-Saharan Africa, where 24 genotypes of the virus have been reported. Between the late 1950s and the early 1980s, genotype I ASFV emerged in Europe, including Italy. In June 2007, a second ASF epidemic wave caused by genotype II was registered, involving several European and extra-European countries, including Italy in 2022. The present paper aims to provide the state of the art of ASF in Italy, describing the course of ASF in wild boars and domestic pigs as an example of multiple concurring different scenarios. Sardinia is coping with the last phase of the eradication of the disease by applying the exit strategy. Conversely, four clusters of infection located in North, Central, and South Italy are still ongoing. The unique and complex Italian experience in ASF-controlling may be useful to increase know-how on the efficacy of strategies and measures, as well as issues that could be further improved.

## 1. Introduction

African swine fever (ASF) is a severe haemorrhagic contagious viral disease of domestic pigs and Eurasian wild boars (*Sus scrofa*) caused by the African swine fever virus (ASFV). ASF can cause huge economic losses in affected pig sectors given its high lethality, which is close to 100% in pig populations, as well as dislocations in pig markets [1]. The ASFV’s transmission occurs in different ways; three main types of cycles have been described so far. The sylvatic cycle is characterised by the circulation of ASFV in wild reservoirs like bush pigs, warthogs, and soft ticks (*Ornithodoros* spp.), which represents the natural hosts of the virus. The domestic cycle occurs when the virus is transmitted among domestic pigs independently of the incursion way. Therefore, biological vectors (*Ornithodoros* spp.) (tick–pig cycle), direct contact (pig–pig–cycle), fomites, contaminated food or water, and other ways that are less likely, such as artificial insemination with contaminated semen or insects that act as mechanic vectors, can contribute to the maintenance of the infection in domestic pigs. Lastly, the epidemiological pattern observed during the current ASF epidemic in Central and Eastern Europe revealed an additional epidemiological cycle involving the Eurasian wild boar (*Sus scrofa*), the wild boar’s habitat, and their carcasses, namely the wild boar–habitat cycle [2,3]. The ASF virus is very stable; it may resist at a wide range of pH values and temperatures (i.e., for years in frozen meat) and to autolysis, remaining infectious for several weeks in carcasses. It is inactivated only by cooking, at very acid and basic pH values, and by specific chemical compounds [4]. Therefore, the ASF virus that is harboured in wild boar carcasses (fresh, decomposed, dry, or frozen) and in the environment may represent a persistent source of infection through its persistence in the infected territories [2]. The first case of ASF was reported in Kenya in 1921. Initially, ASF occurred in sub-Saharan African countries, where it is still endemic, affecting up to 35 African countries and being maintained in an ancient sylvatic cycle in which African wild suids (predominantly warthogs, *Phacochoerus africanus*) and argasid ticks (*Ornithodoros* spp.) represent the natural hosts of the virus [5]. Currently, four different gene regions—p72, p30, p54, and B602L—are commonly used to identify ASFV genotypes. The genotypic differentiation of ASFV into genotypes largely depends on the amplification and sequencing of the variable 3′-end of the B646L gene that encodes p72, the main capsid protein. To date, 24 genotypes of the virus (I–XXIV) have been defined [2,6]. Only five of the twenty-four genotypes are known as pig-adapted viruses (I, II, VIII, IX, and X), and two genotypes out of these five (genotype I and II) were responsible for several incursions in countries outside of Africa [2].

Between the late 1950s and the early 1980s, the genotype I ASFV emerged in Europe, Russia, the Caribbean, and South America. The ASFV was first identified in Europe in 1957, in Portugal, and was then re-introduced in 1960, at which point it quickly spread into Spain, France, Malta, Belgium, Italy, and the Netherlands [7]. In 1978, a new ASF incursion occurred in Sardinia (Italy). The ASFV was also reported in Russia in 1977, and, in the late 1970s, it emerged in Brazil, Cuba, and the Caribbean Islands. By the mid-1990s, the ASFV was eradicated outside of Africa, with the exception of an isolated outbreak in Portugal in 1999, due to its introduction into a shelter of pigs infested by *Ornithodoros erraticus* ticks, and the island of Sardinia (Italy), where it has remained endemic for a long time [7,8]. In June 2007, a second ASF epidemic wave was registered, with genotype II being confirmed in the Republic of Georgia. Subsequently, the virus spread to Armenia [9] and Russia [10]. In 2014, the ASF reached the European Union; in Poland [11], Latvia [12], and Estonia [13], outbreaks of the genotype II in both pig and wild boar were detected, while Lithuania [14] reported outbreaks only in wild boars. In 2017, the ASFV reached the Czech Republic [15] and Romania [16]; the following year, Belgium [17], Hungary [18], and Asian countries [19] reported cases of ASFV. Serbia [20] and Slovakia [21] were reached by the virus in 2019, while Germany [22] and Greece were reached in 2020 [23]. Lastly, in 2022, the ASFV spread to Italy [24], initially involving only wild boars. In the last few decades, wild boar populations underwent a considerable expansion that led to the colonisation of new habitats such as agricultural lands and peri-urban and urban areas. The Italian Apennines showed a high wild boar population, ranging from four to eighteen animals for Km^2^ [25]. A more recent study on wild boar distribution from Western Europe to Central–Northern Asia showed a large variety of animal densities in Italy, ranging from zero to up to more than ten, depending on the characteristics of the territory and its vegetation stratum [26].

The present paper aims to provide the state of the art of ASF in Italy, describing the spread of ASF in wild boars and domestic pigs as an example of multiple concurring different scenarios. Indeed, with regard to ASFV infections in pig and in wild boar, Italy has been facing different realities at the same time:The protection of ASF-free territories with large biodiversity and socio-cultural and economic characteristics;The management of the ongoing ASF epidemic among wild boars in complex geographical contexts and the isolated outbreaks in domestic pigs;The management of ASF eradication in territories where domestic pigs and wild boars coexist with backyard and illegal free-ranging animals and where strong local socio-cultural and economic connotations exist (Sardinia).

The unique and complex Italian experience in ASF-controlling might be useful to increase the know-how on the efficacy of the strategies and measures adopted, including aspects and areas of improvements to combat this disease.

## 2. The Different Italian Scenarios

The current Italian epidemiological scenario (updated to the 11th of September 2023) is reported in Figure 1.

### 2.1. Sardinia: A Step Closer to ASF Eradication

Sardinia has been affected by the ASFV genotype I since 1978, when what were probably food wastes containing ASF were introduced in the region. Although the density of wild boars, ranging from zero to one animal per km^2^, is relatively low compared to other Italian regions [26], the infection became endemic due to the simultaneous occurrence of several geographical, socio-cultural, and farming system characteristics. Sardinia is a complex context where the simultaneous presence of domestic pigs, wild boar, backyard pigs, and illegal free-ranging pigs, which are kept in the inner, mountainous areas of the island where they live in close contact with wild boars, has created the ideal conditions for ASFV endemicity. Illegal free-ranging pig breeding has been a fundamental part of the deep-rooted agropastoral Sardinian traditions for several decades. In such system, pigs were fattened in a small backyard for several months before being slaughtered during a family festival, usually in early winter, or a few sows were retained to produce piglets that were slaughtered at the end of the feeding period and used for traditional dishes, popular in all the Italian regions [27]. However, it soon became clear that free-ranging pigs, backyards with a lack or an insufficient level of biosecurity, and, consequently, uncontrolled pig movements were the most likely causes of the ASF persistence in Sardinia [28,29,30]. Therefore, these peculiar farming systems have represented the key points where eradication measures need to focus on in order to be successful. Therefore, in 2015, a new ASF eradication strategy (EP-ASF-15-18), that was better adapted to the local context, was adopted and then fully implemented by 2016. Such an ad hoc eradication strategy was fully supported and empowered by all local authorities and stakeholders. The eradication plan confirmed the banning of free-range pig keeping (Regional Decree n.69, 18th of December 2012, approved by the Decision 2011/807/UE) and imposed biosecurity regulations on outdoor Sardinian farms. Incentives were provided to farmers to ensure the adoption of biosecurity rules and to abandon illegal practices, while disease-awareness-raising campaigns were also carried out [31]. In more detail, these incentives were funded by the Sardinian region for the implementation of agricultural and rural development programs. The incentives included funding to implement the farming biosecurity level requested, such as the installation of a double fence, and funding for the reproduction sector, such as an individual incentive for breeder pigs. Veterinary inspections and controls were strengthened along the entire pig production chain in an increasingly rigorous manner. Stricter rules were applied to hunting, including safe disposal of wild boar offal. Control measures were accompanied by intensive training, and awareness and communication activities targeted to farmers, hunters, and the population. Outdoor, double-fenced pig farms were authorised and subsidised as an alternative to keeping free-ranging pigs. However, almost 5000 free-ranging pigs had to be culled during around 60 military-type actions starting from November 2015 [32]. These actions allowed Sardinia to remove illegal free-ranging pig farms, reduce the wild boars population, and improve the biosecurity level of all types of farms, as well as reduce the number of new ASF outbreaks in domestic pigs and wild boars [31]. All the above actions were made possible by the implementation of an effective chain of command at the regional level that, with the support of the central authority, applied the technical notions that experts had long suggested.

Thanks to the ad hoc eradication program, the ASFV was no longer detected in Sardinia. The last virus detections date back to 2018 and to 2019 in domestic pigs and wild boars, respectively. Although this result is really important and shows that the joint efforts have been successful, more is needed to obtain the ASF-free status at a European level. In fact, based on the EFSA exit strategy, it is necessary to provide cumulative evidence of the absence of ASFV circulation in wild boar populations [33]. A practical interpretation of the strategy was implemented based on the failure probabilities of wrongly declaring the freedom-status of an area even when the disease is still present but undetected. Therefore, a screening and then a confirmatory phase, complementary to each other, were implemented. The active and passive surveillance of domestic animals, as well as wild boars, was implemented. To increase the number of wild boars investigated and standardise the number of carcasses to be found in a 1000 km^2^ area, the WBC-Counter Tool was developed [34]. Moreover, the active surveillance of wild boars during the hunting season was applied.

The regionalisation criteria provided by the EU regulations were applied in a very strict form in Sardinia: the entire region was considered to be at the highest risk level for a prolonged period of time, notwithstanding the field and laboratory evidence of the presence of active ASFV circulation only in the area around the province of Nuoro. More recently, the restricted zone has been reduced several times, recognising the positive epidemiological evolution.

During the application of the Sardinian exit strategy, ASFV seropositive (Ab+) wild boars and pigs have still been occasionally detected; therefore, the ASF-free status cannot be recognised and declared. However, the decreasing number of detected seropositive animals may suggest that the ASFV eradication in Sardinia may be achieved in the near future.

### 2.2. Piedmont–Liguria–Lombardia Regions: The Challenge of Tri-Regional Incursion

At the beginning of 2022, an initial ASFV of genotype II was reported in wild boars in the north-west of Italy [24]. On the 5th of January 2022, in compliance with the provisions of the national surveillance plan for ASF, the veterinary services collected tissue samples from a wild boar that had been found dead in the municipality of Ovada, in the Alessandria province, a small town in a rough and mountainous area between Piedmont and Liguria. The detection of ASF using molecular methods was performed by the regional animal health laboratory—the Istituto Zooprofilattico Sperimentale del Piemonte, Liguria e Valle d’Aosta (IZSPLV)—and was subsequently confirmed by the National Reference Laboratory (NRL) for ASFV—the Istituto Zooprofilattico dell’Umbria e delle Marche (IZSUM)—on the 6th of January 2022. The real-time PCR test and sequencing analysis showed a circulation of the ASFV genotype II, which was phylogenetically closely related to the strain concurrently circulating in Europe and Asia. In the following days, two other ASFV-positive cases in wild boars in Fraconalto (province of Alessandria, Piedmont region) and Isola del Cantone (province of Genoa, Liguria region), about 30 and 40 km away from the first case, respectively, were detected. Since this was an outbreak in the wild, an emergency meeting of the operational group of experts was organised to define the infected area and outline the extraordinary measures that needed to be implemented in order to limit the spread of the disease. Two regional crisis units (RCU) were established to organise the search for other wild boar carcasses in the territory, checking the pig farms in the infected area, managing the hunting activities, providing operational plans to stakeholders, and implementing any other measures required by the regulations that were necessary to avoid the spread of the disease. In particular, a reinforcement of the passive surveillance in the wild sector and an increase in the vigilance level on biosecurity measures in the domestic sector, with special attention to all the transport and handling operations of animals, feed, animal products, and people, were implemented. At the EU level, in application of the principle of regionalisation and based on the Commission Implementing Regulation (EU) 2021/605, which was applicable at the time, the restricted zones I and II were defined and the restrictions on suids and swine products’ movements in these areas were applied to prevent the spread of the disease out of the restricted zones, which comprised the surveillance zones and the infected zones, respectively, in this case where the outbreaks of ASF were in wild boars.

To achieve the best results in a complex bi-regional scenario, with the decree of the President of the Council of Ministers of the 25th of February 2022, a special Commissioner was appointed to carry out the task of coordinating and monitoring of the actions and measures put in place and, when necessary, to be able to issue ordinances with an immediate effect, in order to prevent and contain the spread of ASF. Among this authority’s decisions, an estimation of the local wild boar density (high in this area, ranging from 60,000 in Liguria to 85,000 in Piedmont) and an accurate and detailed study of the territory affected by the ASF were conducted to define which area to fence in order to stop the advance of ASF towards free territories, following the previous, successful management of ASF by Belgium. However, Italian bureaucracy slowed down the building of the fences, and, by the time construction was completed, some positive cases had already been found outside the fenced perimeter. Moreover, the particular landscape involving the urban areas, which were crossed by torrents and bridges and surrounded by rural and forest areas, made it difficult to completely close the whole perimeter, and a small stretch of fence was never constructed. Therefore, despite the measures put in place, the ASF continued to spread, and new positive cases in wild boars were detected in several other municipalities. On the 26th of June 2023, the first positive case in wild boar in the Lombardia region was confirmed, making this a tri-regional incursion. From June to August, more and more cases were confirmed in wild boar, and the first positive outbreak in pigs was detected on the 18th of August 2023. To date, ASFV has involved eight big pig farms in the Lombardia region. Protection and surveillance zones were promptly defined, and culling and disposal of pigs was implemented to eradicate the outbreaks. Lombardia, like other north-Italian regions, is characterised by an intensive farming system, with an average of over 1000 animals per farm. Although a high biosecurity level for swine farms is required by the Italian regulation (decree from the 28th of June 2022 on the biosecurity requirements for swine farms), the official verification of the affected farms by the veterinary local authorities during veterinary inspections showed the presence of gaps such as the lack of disinfection procedures at farm entrances. A deep cooperation among local, regional, and central authorities, veterinarians, and farmers has been requested in order to control this alarming epidemiological situation in North Italy.

### 2.3. Central Italy, Lazio Region: A Re-Emerging Challenge

Lazio is a central Italian region that is characterised by the high wild boar population (around 70,000 animals) living in rural, as well as in peri-urban and urban areas, and a poorly pigmeat sector that is characterised by backyard pig farming and commercial farming, typically attaining no more than 50 animals.

The ASFV was detected in a wild boar in the north-west of the city of Rome on the 5th of May 2022. Similarly to the outbreak in North Italy, prompt notification of the competent EU and international authorities and a careful study of the territory were performed to identify the most effective measures for preventing the spread of the disease. The Lazio region established the core of the infected area within the highway named “Grande Raccordo Anulare”, surrounding the city of Rome, but the restriction zones included part of the Insugherata Park, Veio Park, Pineto Park, and Monte Mario Reserve. To confine the infection, fences were installed to close the gaps where wild boars could exit from the infected area and spread the ASFV in the peripheral free zones. Moreover, the following measures were taken: the reinforcement of the passive surveillance carried out by the personnel of “Roma Natura” and the veterinary services on wild boars; the disposal of wild boar carcasses following strict biosecurity procedures; the installation of specific signage to delimit the areas involved in ASF cases; the prohibition of feeding, approaching, and disturbing wild boars, as well as prohibiting the organisation of events and outdoor activities such as picnics, trekking, mushroom picking, etc.; and the fencing of garbage containers to prevent access by wild boars. Additionally, an estimation of the wild boar population (calculated to be around two–three animals for km^2^), a census of the commercial and family farms raising pigs with an update of the National Database, and the verification of the presence of illegal free-ranging pigs or pigs housed for non-commercial purposes (pigs as pets) were also performed. Over the following months, the number of positive carcasses that were locally found increased. Moreover, a suspected case of ASF was also reported in a wild boar accident in the Province of Rieti (about 90 km from the discovery in Rome). The NRL confirmed the presence of the ASFV, and the case was reported to the EC. The surveillance activities carried out for over five months did not detect any positive carcasses and made it possible to exclude the viral circulation in this area. Thus, the suspected case was downgraded, the restriction zones I and II were removed, and the free status of the province of Rieti was promptly restored.

Unfortunately, despite the measures adopted in the infected area of Rome, an outdoor pig farm was found to be ASF-positive on the 9th of June 2022, not far from the area where the first ASFV-infected wild boar carcass in the region had been discovered. Following the emergency plan provisions, the following actions were performed: the restrictions on animal and animal products’ movements from the infected farm; the establishment of the protection and surveillance zones; the stamping out of the infected farm (epidemiological unit); and the epidemiological enquiry. The lack of farm biosecurity in a suburban environment that was highly populated by wild boars was hypothesised to be a major risk factor for ASF introduction into the domestic pig farm. The disposal of carcasses by applying strict biosecurity procedures and the cleansing and disinfection of the involved farm were performed, and the eradication of the outbreak was promptly achieved; unfortunately, especially considering the poor quality and inappropriate facilities, operations were not conducted optimally. In the restriction zones II and I, the ASF-positive cases in wild boars increased until September 2022, when the growth stopped. From September 2022 to May 2023, no more new cases in wild boars were detected, and the authorities believed that the eradication of the outbreak had been successful. However, new cases were detected again in May, and a new epidemic phase is occurring (Figure 2).

### 2.4. Calabria Region: A Complex Scenario

Calabria is a southern-Italian region characterised by the moderate wild boar population (around 60,000 animals) in rural and forest areas and a small pigmeat sector that is mostly characterised by familiar or wild and semi-free-range farming, typically attaining no more than 200 animals.

On May 2023, three wild boars that had been found dead in the municipality of Cardeto (Reggio Calabria province) were confirmed to be positive for ASFV by the NRL. Then, another positive case was found in the municipality of Bagnara Calabra, 40 km far from Cardeto, where the first case in the region had been recorded. In this territory, the wild boar population appears moderate, ranging from zero to two animals for km^2^; however, the presence of several backyard pigs and small commercial farms with low biosecurity levels represents a risk factor for ASFV introduction. Therefore, the reinforcement of passive surveillance of wild boars, and the implementation of biosecurity measures in infected zones and surveillance zones were applied to avoid the spreading of the disease in neighbouring, free zones. Later, the infection spread to two small herds of domestic pigs in the municipality of Africo, 70 km far from the previously identified outbreak in wild boars. The herds were located in a rural forest area in the Aspromonte National Park and no biosecurity measures were in place. Protection and surveillance zones were defined for the outbreak, and the stamping out of around 80 animals was performed. Properly burying carcasses on the farm, and cleaning and disinfection procedures were carried out to eradicate the outbreak promptly.

Despite the biosecurity measures put in place, new cases in wild boars occurred and the infection is still spreading in all directions.

The situation immediately appeared rather difficult to manage; the pig registry of the affected farms was found to be out-of-date, and the same was true for other farms in the area. The general biosecurity level of the farms appeared deficient, and it is certainly safe to assume that many animal movements may have escaped veterinary control. The wild pig population was likewise difficult to control, both because the territory was very rugged and wide, and because the regional authorities struggled to organise a systematic tracing of carcasses at least at the edge of the infected area, as indicated in the national contingency plan.

### 2.5. Campania Region: Newest Outbreak

Campania is a central-southern Italian region that is characterised by the moderate wild boar population (around 55,000 animals) in rural and peri-urban areas and a poorly pigmeat sector that is mostly characterised by familiar or wild and semi-free-range farming, typically attaining no more than 20 animals; scattered commercial farming with more than 100 animals can also be found.

On the 22nd of May 2023, the disease was confirmed in Campania, in the municipality of Sanza and Montesano sulla Marcellana (Salerno province), on five wild boar carcasses in an advanced state of decomposition. The President of the region signed ordinance n. 1 on the 26th of May 2023 establishing the infected zone in the municipalities of Buonabitacolo, Casalbuono, Casaletto Spartano, Castelle in Pittari, Montesano sulla Marcellana, Monte San Giacomo, Morigerati, Padula, Piaggine, Rofrano, Sala Consilina, Sassano, Sanza, Teggiano, Torraca, Tortorella, and Valle dell’Angelo. However, considering the other recent positive carcasses that were found close to the border with the Basilicata region, the outbreak management in wild boars has required a bi-regional involvement. An estimation of wild boar density showed one to five animals for km^2^. To prevent the further spreading of the ASFV in the territory and the involvement of the domestic pig sector, a coordinated and efficacious campaign to reduce the wild boar population by using targeted hunting and trapping with euthanasia was started.

## 3. Discussion

African swine fever (ASF) is a highly fatal disease of domestic pigs and wild boars. It is a threat to the pig industry as it lowers production and significantly impacts livelihoods [35]. Although research on vaccine development has been ongoing for some time, neither a vaccine nor a therapeutic product for ASF are currently available, making the control of the disease challenging. Therefore, the impact of ASF, once introduced, can only be minimised through the adoption of strict biosecurity measures [35] and effective preparedness planning. Considering the international scenario characterised by several incursions of ASF in several European and non-European countries, since 2018, Italy has put in place some actions that are necessary to prevent ASF introduction in peninsular Italy, as well asbe ready to face the emergency when an ASF incursion occurs. Therefore, the Italian contingency plan, the emergency manual for ASF in wild boars, and the operative manual for ASF and classical swine fever (CSF) in domestic pigs were drafted. Awareness campaigns for all stakeholders and field or tabletop simulation exercises on outbreak coordination and crisis response were carried out. Such activities involving public health authorities and veterinary services were performed to implement the areas of preparedness and response planning, monitoring, surveillance, crisis management, and risk and crisis communication. However, despite the efforts to avoid the incursion of ASF in Italy, in 2022, the genotype II ASF was reported in six different peninsular regions, while the Sardinia region was applying the exit strategy to obtain the ASF-free status from the EC.

The first incursion in the Italian peninsula was on the 5th of January 2022 in the Piedmont region. Here, as in all the other following outbreaks, the ASFV infection was first identified in wild boars. The control of ASF in wild boars requires an inclusive strategy involving various measures. In the early stages of the emergency, Italy responded in an effective and timely manner, adopting measures approved by the EC. Through the guidance of international experts who had gained direct experience with ASF in other contexts, an onerous but viable strategy was devised and, above all, it had a good chance of success. One critical issue in the practicality of the strategy adopted was the not exactly simple institutional dialogue between central authorities/national experts and regional institutions, and, consequently, the low engagement of the local population in accepting and respecting the restriction measures adopted, such as the ban of public and social events in areas under restriction. This social factor certainly complicated the management of ASF control measures and their sustainability. The construction of a fence system to prevent the ASF spreading from the infected zone to the free areas, the quick removal of the infected wild boar carcasses in the infected area, and the decrease of the wild boar population in the buffer area surrounding the infected area were the most important actions implemented. With regard to the fencing system, a modified and reduced fence project was prepared and constructed with delays, allowing the infection to spread outside before its completion. This proved that the Italian administrative procedures necessary to construct the fencing system were a critical issue for the success of the control strategy.

To date, around 650 confirmed cases of ASF have been reported in North Italy (Piedmont, Liguria, and Lombardia), and eight domestic outbreaks were detected in the last month, producing a considerable economic loss in Italian pigmeat in a short time.

The ASF is continuing to spread in peninsular Italy, involving additional new areas and, lastly, the first outbreaks in domestic pigs have occurred. Although the wild boar carcass-persistence time and the ASFV-persistence in the soil underneath infected carcasses are still two main sources of uncertainty in ASF epidemiology, fast localisation and removal of carcasses are considered one of the most important disease-control measures in the affected regions [36,37,38]. For this reason, the principal strategy of the Italian surveillance and eradication programme for African swine fever is the strengthened passive surveillance of wild boars in the restriction zones I and II. This activity is based on the active search for wild boar carcasses on the Italian territory and their safe disposal. Recently, the Italian government, through the ad hoc Extraordinary Commissioner for ASF, is suggesting a strategy based on the depopulation of wild boar from unsuitable areas, such as urban and peri-urban areas. To securely carry out these actions, biosecurity training has been arranged to give specific information on biosafety matters to the hunters who will assume the role of “bioregulators” and will be registered in an official national list. Furthermore, passive surveillance of domestic pigs (molecular test on the spleen of dead animals; two pigs/week per region) is being carried out throughout the entire Italian territory. Moreover, official veterinary verification of the biosecurity levels applied in the farms has been intensified, and active surveillance has been implemented. In particular, without prejudice to the specific prohibitions in the protection and surveillance zones established in the Commission Delegated Regulation (EU) 2020/687 and in the restriction zones according to the Commission Implementing Regulation (EU) 2023/594, the movements of pigs, both for life and for slaughter, from the farms in the regions with restriction zones for ASF, as well as in those epidemiologically linked to the regions with domestic outbreaks, can only take place in the following cases: (1) The clinical visit, performed in the 24 h before the first load, does not show any suspected clinical signs. The clinic visit must be repeated every 72 h for the following loads; (2) The verification of mortality trends, particularly in the 24 h preceding the first load, does not show any anomalies. Moreover, the regions and autonomous provinces can perform a molecular test on spleen samples in dead animals or blood samples in non-healthy animals in the 72 h preceding the first load, based on the epidemiological situation and assessment of risk.

Summarising, the eradication plan presented by Italy in 2022 for the north-west cluster was, in practice, not applied in its essential parts. In particular, the effective containment of the infected population through the double barrier system and the creation of a buffer in which to apply depopulation were lacking. With different approaches and results, the regional authorities applied an active search for carcasses which representing a persistent source of infection for other animals. Precisely in this perspective, the ASF epidemic in Italy presents unique characteristics, such as the virus circulation in the context of a Mediterranean scrub with a high density of wild boars and a very peculiar geographic conformation. In this view, the only similar experience in controlling ASF is represented by Sardinia with genotype I. However, in Sardinia, the primary risk factors were represented by a high number of free-ranging pigs and uncontrolled pig movements among them. While these risk factors are not present in North Italy (Piedmont, Liguria, and Lombardia), the human factor represents one of the most significant risk factors in this area. Indeed, human activities, such as the collection of wood and other forest products, outdoor tourism, hunting, etc., create the basis for the easy and uncontrollable spread of the disease, although no evidence is available on the link between the infection clusters in North, Central, and South Italy. The NRL and the National Reference Laboratory for Gene Sequences are studying Italian isolates to try to find related ones; however, the phylogenetic analysis of the whole genome sequences requires good quality samples, time, and economic resources, and is not even easy to achieve.

While a local endemic persistence of the ASFV and a steady geographic spread to neighbouring disease-free areas, like a spot of oil, has occurred in Piedmont–Liguria–Lombardia, ASF has been making long-distance jumps involving Lazio, then Calabria, and, in the end, Campania. As it is known, ASF can easily spread by close contact between infected wild boar and healthy animals, especially when the pressure of the infected wild boars in the infected area and the population of wild boars in the ASF-free area are both high. This is what occurs in the Italian infected zones, such as Piedmont–Liguria–Lombardia, where local endemic persistence of the ASFV implies a steady geographic spread, like a spot of oil, to neighbouring disease-free areas. However, ASF is able to jump and cover long distances too. Human activities, such as the illegal trade of animals or pork products, play a key role in involving territories that are far from the infected zone. In Italy, ASF long-distance jumps occurred when the virus spread from North Italy to Lazio (around 500 km) and then Calabria (around 800 km). The epidemiological investigations performed on the outbreak in wild boar were not useful in detecting the exact source of the infection, and it was only possible to speculate. As demonstrated by previous ASF outbreaks globally, when the disease is detected in wild boars, sooner or later, it is reported in domestic pigs as well, as occurred in Italy. Indeed, outbreaks in domestic pigs occurred in both Lazio and Calabria, and, lastly, in the Lombardia region. In South Italy, every single domestic outbreak had already been eradicated. All these cases were linked to small pig farms with semi-wild breeding, with poor or lacking biosecurity measures, where contact between wild boars and pigs could occur easily. In particular, in the Lazio region, after the eradication of the outbreak in domestic pigs, there was a prolonged period (from September 2022 to April 2023) with no more positive cases being detected in wild boars. The epidemiological silence evidenced in the municipality of Rome for several months had given all the experts hope. The containment ensured by the highway that runs around the city, the small size of the park in which the infected wild boar population resided, and the constant monitoring carried out by local institutions represented strong premises for the eradication of the cluster infection. Instead, new cases were reported in May and June 2023. The incidence of ASF-positive cases in wild boars became very high again, and a new epidemic phase occurred. The origin of the new epidemic wave in Central Italy is not yet clear; a new incursion or a rising infection rate over the detection threshold are the major hypotheses.

Currently, the detection of some cases outside the “Grande Raccordo Anulare” is of great concern because they could represent the beginning of the expansion of the infection towards an area without natural and/or artificial barriers. This high-risk situation has been made even more complex by the intervention of animal rights activists who oppose the ASF control measures adopted. In particular, they sabotage the traps installed for depopulation in the infected area or “rescue” sick animals by removing them from the control of the veterinary service. The latter activity is particularly risky because it promotes the spread of the infection through the movement of animals that could be infected with the ASFV.

In Calabria, the detection of ASF cases in wild boars and, subsequently, the outbreak in domestic animals challenged the regional system. Although several training courses, involving both veterinarians and other stakeholders, were held recently, the practical level of awareness in face of the facts was not found to be adequate to deal with the emergency. In particular, shortcomings have been highlighted in the management of informative systems and information flows, which are also the basis of epidemiological investigations. It seems clear that the endemic trend of ASF in this area is already consolidated and, therefore, it will be necessary to carefully study the epidemiological evolution of the infection in this context.

On the contrary, the reaction of the veterinary system in the cluster identified in the Salerno Province was more effective; the actual infected area appears to be quite limited at the moment, but the density of the wild boar population and the conformation of the territory worry the experts, who fear a probable evolution in an endemic sense in this case as well.

In this context of new incursions of the virus, the history of ASF in Sardinia is completely separate. Sardinia has been facing the ASFV since 1978. Thanks to the last Sardinian eradication plan of 2015–2018 (ASF-EP15/18), based on banning free-range pig keeping (Regional Decree n.69, 18th of December 2012, approved by Decision 2011/807/UE), which was considered to be the principle cause of ASF endemicity, and on strict biosecurity regulations on outdoor Sardinian farms, a marked improvement of the epidemiological scenario was achieved. The number of outbreaks in both wild boar and domestic pig populations decreased in the following years. In September 2018, the virus was detected for the last time in domestic pigs, while the last PCR-positive illegal free-ranging pigs and wild boars were detected in January 2019 and April 2019, respectively [31]. Recently, the EFSA proposed a strategy that is necessary in the last phase of ASF eradication to provide robust evidence of the absence of the ASFV circulation in wild boar, called exit strategy. This approach is primarily based on the passive surveillance of wild boar using molecular and serological tests. Since, according to the Regulation (EU) 2016/429 of the European Parliament and of the Council, the seropositive animals are considered confirmed positive cases and, therefore, reportable to the EC, the success of the exit strategy is strictly dependent on the lack of seropositive animals in the investigated territory. The exit strategy is characterised by two phases: a screening phase, at first, and, then, a confirmatory phase. Each country must elaborate its specific exit strategy tailored to the epidemiological context, the time necessary to complete the screening and confirmatory phases, and estimate the number of carcasses to be found in order to demonstrate the free status. Therefore, Sardinia developed a well-functioning specific exit strategy and a decrease in seropositive animals was observed. However, some scattered seropositive animals (both in domestic pigs and wild boars) have still been detected, so the ASF-free status still cannot be requested to the EC. In this respect, it should be noted that the seropositive animals still found in Sardinia represent a reflection of the past active viral circulation in the territory and not an indicator of active viral circulation. From an epidemiological point of view, the finding of a low number of seropositive animals in the context of active surveillance has no significance. Indeed, several authors have downsized the role of seropositive animals as virus shedders in both domestic pigs and wild boars [39,40]. Based on these data, an Italian request to the EC to eliminate the serological surveillance in the last phase of eradication in Sardinia is ongoing. This step is considered necessary to speed up the process of obtaining ASF-free status.

## 4. Conclusions

In Italy, there is a complex scenario characterised by the management of the exit strategy for the ASF genotype I eradication in Sardinia and four new incursions in peninsular Italy by the ASF genotype II. Although ASF has been spreading in several European countries, the current Italian scenario, with four distinct ASFV incursions, could not be foreseen. The approach adopted in Italy to control and eradicate the disease is similar to that applied in several other countries and follows the EFSA and FAO guidelines. However, the success depends on how stringently and effectively the implemented measures are imposed by each country. Past experiences showed that countries like Belgium and Czech Republic were able to cope with the ASF challenge and eradicate the disease, while other countries (such as Italy, Germany, Poland, Romania, Hungary, Estonia, Latvia, Lithuania, etc.), despite the implemented measures, are having difficulties stopping the spread of the disease. ASF is a very challenging disease that reflects a complex interaction of sanitary, economic, environmental, political, sociological, and cultural factors. These factors form an intricate meshwork where several stakeholders belonging to the national, regional, and local authorities, farmers, hunters, forest rangers, and citizens have to communicate and collaborate together to combat ASF effectively. Moreover, ASF requires expensive control measures in both the preparedness and eradication phases that have to be implemented in a timely manner. The experience gained in Piedmont regarding the fence construction suggests that improvements are necessary with regard to the implementation of measures identified, and it highlighted the importance of all stakeholders’ agreement on the strategy outlined. There was probably a communication failure, but certainly many conflicts of interest have had a bad influence. The great significance that the problem attributed to the overpopulation of the wild boar and the damages that this causes, especially to agriculture, has created a media effect in the country that often overlapped with the emergency of the ASF. Someone thought it possible to solve the two problems simultaneously or to take advantage of one to tackle the other, but often there was just confusion. Actually, in order to eradicate ASF in wild boars, it is necessary to prepare specific intervention (eradication) plans that are linked to the territory in which the infection is present. In infection-free territories, reducing wild boars’ density helps in terms of ASF prevention but also remains a major benefit for farmers. The tendency towards endemic in the Italian clusters and the progressive expansion of the infected area in the north-west poses the problem of how to defend the productive areas of the country. Biosecurity remains the central pivot on which to work, especially with regard to intensive farming, not only in terms of structures, but also in the mentality of the operators. Moreover, strict control of the entire supply chain is necessary to offer adequate guarantees, especially for exports. However, there are many districts where pig farming is based on backyard or extensive farms in Italy; therefore, a sustainable system for this type of chain will have to be quickly found.

As known, ASF is not a zoonosis and, therefore, a strict One-Health approach cannot be considered necessary. However, ASF causes direct and indirect economic impact, and, when investments are to be moved to this disease to cope with its emergence, other important sectors like the healthcare sector may be subject to underfunding with a reduced capacity to maintain the level of care of the national health system. For this reason, this disease needs a change of mindset; the regional investments in the animal health system infrastructure, biosecurity, and implementation of surveillance capacity, and the ability to actively involve all the stakeholders, must be considered a priority in the same manner as the actions that are put in place for the diseases that require the One-Health approach.

## Figures and Tables

**Figure 1 animals-13-02998-f001:**
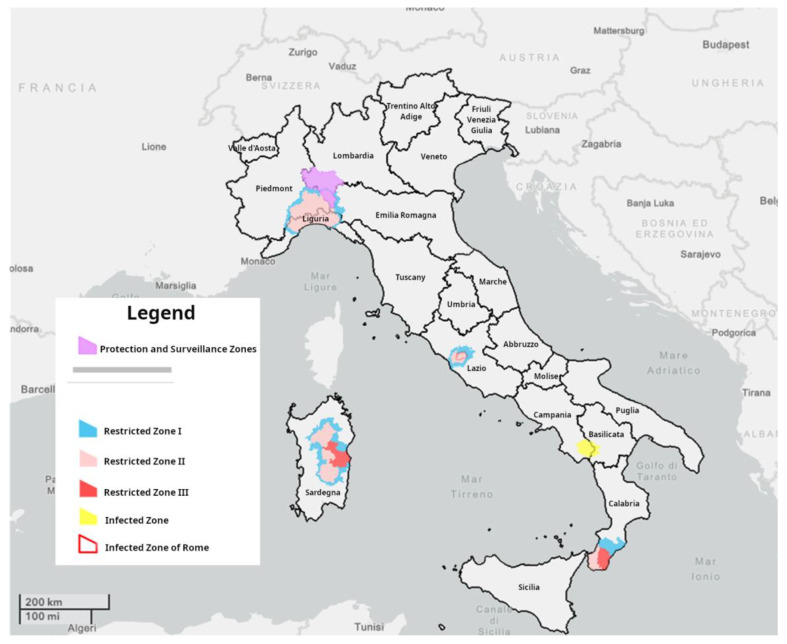
Current Italian epidemiological scenario updated to the 11th of September 2023 representing the protection and surveillance zone and the restricted zones I, II, and III that are listed in the Annex I of Commission Implementing Regulation (EU) 2023/594, as last amended by the Commission Implementing Regulation (EU) 2023/1485 on the 18th of July 2023.

**Figure 2 animals-13-02998-f002:**
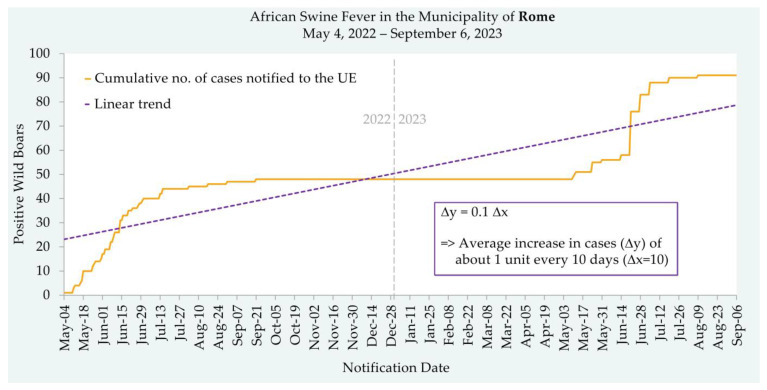
Daily trend of the cumulative number of ASF cases reported to the EU from the 4th of May 2022 to the 19th of July 2023 in the Rome municipality.

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
