# Peer review of "African Swine Fever: The State of the Art in Italy"

_animals, 2023, doi:10.3390/ani13192998_

Round 1

Reviewer 1 Report

Dear authors

Thank you for an interesting an honest manuscript describing the management of ASF in different regions of Italy. Although, I find the subject interesting, in my opinion, this is not a scientific review. And there is nothing "state-of-the-art" in this in this. 

I think the manuscript would benefit from shortening quite a bit, and adding some region names or similar on the map. And to re-submit as another type of publication.

A few details:
Your use of outbreaks is not the same as defined by woah. I guess you define outbreaks as the epidemics running in different regions, but this is not clear.

Chenais et al described four cycles - on is called "habitat" - not "habit"

Virus from carcasses is probably not a "constant" source of infection. But it clearly has an impact on the spread.

It is a bit odd that you mention Asia before EU, as to my knowledge the first introduction in Asia was in 2018?

You mention in Sardinia "incentives to respect biosecurity" - it would be helpfull for others to know which these incentives were, since they seem to work.

Piedmont-Liguria: Please shorten the first part about testing.

-

Author Response

Reviewer #1:

Comments and Suggestions for Authors

Dear authors

Thank you for an interesting an honest manuscript describing the management of ASF in different regions of Italy. Although, I find the subject interesting, in my opinion, this is not a scientific review. And there is nothing "state-of-the-art" in this in this.

I am glad that the manuscript was appreciated as an honest contribution to the epidemiology of ASF. We made the changes following the suggestions received to satisfy the requests of all reviewers. Considering the request of the 1° and 3° reviewers about the figures, we updated the images (and text accordingly) with the new recent epidemiological data, improving their resolution.

I think the manuscript would benefit from shortening quite a bit, and adding some region names or similar on the map. And to re-submit as another type of publication.

A slight shortening of the text has now been performed; however, the other requested reviewers' additions to the text have stretched the paper again.

The length of the manuscript does not fit with “short communication”. Therefore, we ask the Editor the decision about the type of publication we must use to publish the present paper.

A few details:

Your use of outbreaks is not the same as defined by woah. I guess you define outbreaks as the epidemics running in different regions, but this is not clear.

Thank you very much for your remark because a deep check of the text has now been performed and a revision of the improper use of outbreak was made according to the definition of outbreak indicated by WOAH, i.e. “an occurrence of one or more cases in an epidemiological unit”.

Chenais et al described four cycles - on is called "habitat" - not "habit"

                The suggestion has been received.

Virus from carcasses is probably not a "constant" source of infection. But it clearly has an impact on the spread.

                The suggestion has been received and the text modified accordingly.

It is a bit odd that you mention Asia before EU, as to my knowledge the first introduction in Asia was in 2018?

                The text was modified accordingly.

You mention in Sardinia "incentives to respect biosecurity" - it would be helpfull for others to know which these incentives were, since they seem to work.

More detailed information were added in the text.

Piedmont-Liguria: Please shorten the first part about testing.

The paragraph has been mildly shortened; however the new epidemiological scenario in North Italy with outbreaks in domestic pig has been requested an updated of the manuscript and new information has been added stretching a little bit again the text.

Reviewer 2 Report

The manuscript “African Swine Fever: state-of-the art in Italy and future scenario” describes the current epidemiological situation regarding ASFV in Italy. The paper includes also a historical aspect of the long-lasting efforts to eradicate the disease in Sardinia, which has been endemic up to 2019.

ASF is currently the most important threat to the swine industry worldwide, and several countries has faced challenging and complex task of the disease control. The paper describes unique experience of Italy, both the example of Sardinia and recent introductions of ASFV into several different regions, and provides an opportunity to study the outcomes of different tools and control scenarios. Considering the current epidemiological situation regarding ASF, it contributes important information to the general knowledge on the ASF eradication. 

 The manuscript is generally well written and properly organized. The authors describe in details main activities and problems in ASF control in different Italy regions. However, very scarce background information on wild boar population, or a structure of pig production sector in the country were included. Also, it would be interesting to have some more info on the scale and organization of passive and active surveillance.

The authors focus mainly on current problems and obstacles in ASF control, and not much discussion on future scenarios is included, contrary to what the title indicates. I suggest rephrasing the title accordingly.

Some minor comments:

Line 51 – the cited paper (2) does not include indicated information regarding wild boar habit cycle

Lines 81 – 84 This sentence suggest that all the outbreaks took place in 2014, except Italy in 2022. Also, while describing ASF in the EU the authors mentioned wild boars only. Please rephrase.

Line 396, Should be May and June 2023 instead 2022

Lines 469-470 Poland is mentioned twice

Figure 1. It would be helpful to mark the names of specific regions on the map to make it more compatible with the text.

Author Response

Reviewer #2:

Comments and Suggestions for Authors

The manuscript “African Swine Fever: state-of-the art in Italy and future scenario” describes the current epidemiological situation regarding ASFV in Italy. The paper includes also a historical aspect of the long-lasting efforts to eradicate the disease in Sardinia, which has been endemic up to 2019.

ASF is currently the most important threat to the swine industry worldwide, and several countries has faced challenging and complex task of the disease control. The paper describes unique experience of Italy, both the example of Sardinia and recent introductions of ASFV into several different regions, and provides an opportunity to study the outcomes of different tools and control scenarios. Considering the current epidemiological situation regarding ASF, it contributes important information to the general knowledge on the ASF eradication. 

The manuscript is generally well written and properly organized. The authors describe in details main activities and problems in ASF control in different Italy regions. However, very scarce background information on wild boar population, or a structure of pig production sector in the country were included. Also, it would be interesting to have some more info on the scale and organization of passive and active surveillance.

I am glad that the manuscript was appreciated as a general contribution on epidemiological know-how of ASF. We made the changes following the suggestions received to satisfy the requests of all reviewers. Considering the request of the 1° and 3° reviewers about the figures, we updated the images (and text accordingly) with the new recent epidemiological data, improving their resolution.

Background information on wild boar population, structure of pig production sector, and organization of passive and active surveillance have now been added to the text.

The authors focus mainly on current problems and obstacles in ASF control, and not much discussion on future scenarios is included, contrary to what the title indicates. I suggest rephrasing the title accordingly.

The title was modified accordingly. 

Some minor comments:

Line 51 – the cited paper (2) does not include indicated information regarding wild boar habit cycle

            The correct reference has now be added in the text.

Lines 81 – 84 This sentence suggest that all the outbreaks took place in 2014, except Italy in 2022. Also, while describing ASF in the EU the authors mentioned wild boars only. Please rephrase.

            The text was changed accordingly.

Line 396, Should be May and June 2023 instead 2022

            The suggestion has been received.

Lines 469-470 Poland is mentioned twice

            The suggestion has been received.

Figure 1. It would be helpful to mark the names of specific regions on the map to make it more compatible with the text.

            The figure has now been modified as suggested and updated to the current scenario.

Reviewer 3 Report

This manuscript titled “African Swine Fever Virus Transmission and Control: The Role of Wild and Domestic Suids” aims to provide a state of the art of ASF in Italy, describing the course of ASF in wild boars and domestic pigs as an example of multiple concurrent different scenarios.The manuscript is well-organized and has certain significance. It was addressed a specific gap in the field, the references are appropriate, However, there are some problems in this manuscript that need to be revised;

1 The language needs considerable attention.

2 Figure 1 and figure2 are difficult to check. Size is too small. Or can’t see clearly.

Author Response

Reviewer #3:

This manuscript titled “African Swine Fever Virus Transmission and Control: The Role of Wild and Domestic Suids” aims to provide a state of the art of ASF in Italy, describing the course of ASF in wild boars and domestic pigs as an example of multiple concurrent different scenarios.The manuscript is well-organized and has certain significance. It was addressed a specific gap in the field, the references are appropriate, However, there are some problems in this manuscript that need to be revised;

I am glad that the manuscript was appreciated as a general contribution on epidemiological know-how of ASF. We made the changes following the suggestions received to satisfy the requests of all reviewers.

1 The language needs considerable attention.

Thank you for your suggestions. An expert in English revised the text.

2 Figure 1 and figure2 are difficult to check. Size is too small. Or can’t see clearly.

We updated the images (and text accordingly) with the new recent epidemiological data, improving their resolution. The figure are now characterized by 146x120 mm in size/300 dpi/.tiff format and 1400x700 mm in size/150 dpi/.jpg

Round 2

Reviewer 3 Report

good